# Does Fiscal Decentralization Affect Regional High-Quality Development by Changing Peoples' Livelihood Expenditure Preferences: Provincial Evidence from China

**Dingqing Wang** [1] , **Enqi Zhang** [2] **and Hongwei Liao** [3,*]

1  School of Economics, Jilin University, Changchun 130012, China
2  Krieger School of Arts and Sciences, Johns Hopkins University, Washington, DC 20036, USA
3  Center for China Public Sector Economy Research, Jilin University, Changchun 130012, China
*  Correspondence: liaohw@jlu.edu.cn

**Abstract:** The reform of the fiscal and taxation system is important for building a sound livelihood protection system as well as resisting the impact of uncertain events and thus promoting the quality of regional development. We explore the strengths and limitations of China's fiscal decentralization system from the perspective of peoples' livelihood expenditures, and provide an empirical basis for institution building for countries to withstand the shocks of uncertain events and promote high-quality regional development, using each provincial-level region in China as the research object. We find that fiscal decentralization has an inverted U-shaped relationship with regional high-quality development, and the inverted U-shaped relationship of fiscal decentralization with regional quality development is significant in the innovation, greenness, and openness dimensions. It is further found that in the process of constructing regional high-quality development, fiscal decentralization will raise the preference for healthcare expenditures, improve the modern public health system, and indirectly promote regional high-quality development, while it will lower the preference for social security employment expenditure, neglect the basic resident social security employment problem, and slow down the process of high-quality development. This paper expands the research on the correlation between fiscal decentralization, livelihood expenditure preferences, and regional development quality, and provides an important theoretical and practical basis for the improvement of the fiscal system and the improvement of social welfare levels in the post-COVID-19 era.

**Keywords:** fiscal decentralization; high-quality development; healthcare expenditure preference; social security employment expenditure preference; inverted U-shaped; design of policies in the post-COVID-19 era



## 1. Introduction

The COVID-19 pandemic has spurred significant changes in the fields of economic development, social issues, and everyday life [1]. As of April 2022, the COVID-19 pandemic has swept the world for nearly three years, with more than 500 million infections and nearly 15 million deaths worldwide, and public health systems in countries around the world have been severely challenged. Among them, China's public healthcare system has withstood the test, and under the guarantee of the Chinese government's strong execution and perfect healthcare infrastructure, domestic epidemic prevention and control have been effective, and the quality of regional development has been steadily improved. The impact of public health events such as the COVID-19 pandemic on human health and life safety is catastrophic, and the uncertainty of economic development is far-reaching [2]. Improving the existing fiscal system, formulating sound fiscal policies, and choosing the right fiscal instruments can improve healthcare coverage and accelerate economic recovery [3], enhancing public wellbeing while building healthier lifestyles [4]. The resulting reflection on the public health care system and regional development quality increases the value of

our inquiry into the correlation between the government's fiscal system, livelihood-based expenditure preferences, and regional high-quality development. In the post-COVID-19 era, how to improve the fiscal decentralization system, strengthen livelihood infrastructure, guide citizens to build a healthier lifestyle, and increase public wellbeing and thus achieve high-quality development in line with the new development concept has become the key to constructing blueprints for economic and social development in countries now and in the future.

In 2015, the 193 member states of the United Nations officially adopted 17 Sustainable Development Goals (SDGs), which aim to thoroughly address economic, social, and environmental development issues in an integrated manner and shift to a sustainable development path, which is an important way to achieve high-quality economic and social development [5], as well as to enhance national economic resilience and resist public health shocks [6]. In the post-COVID-19 era, countries have constructed evaluation systems for high-quality development based on the UN Sustainable Development Goals (SDGs), including the US New Economy evaluation system [7], the EU Sustainable Development Evaluation System [8], and the German national welfare evaluation system [9], and the improvement in the quality development indicator system reflects the importance they attach to sustainable development. From the practical history of China and the development experience of other countries around the world, the measurement of development quality can affect the realization of development concepts and development goals, and a corresponding measurement index system must be constructed to promote high-quality regional development [10]. The Chinese government has also given an official definition of high-quality development: "High-quality development is development that can well meet the people's growing needs for a better life, development that reflects the new development concept, development in which innovation becomes the first driving force, coordination becomes an endogenous feature, greenness becomes the universal form, openness becomes the necessary path, and sharing becomes the fundamental purpose". https://theory.gmw.cn/2020-11/10/content_34354489.htm (accessed on 8 August 2022). Therefore, it is important to construct a system of indicators for high-quality development in China based on the new development concept. High-quality development is affected by the political environment [11], cultural and legal environment [12], human capital quality [13], infrastructure construction of the urban living environment [14], and environmental quality [15]. To break down the barriers to high-quality development, technological innovation, participation of social organizations, and effective government system supply will become powerful weapons [16–19]. In the post-COVID-19 era, the government, as the strategic planner of regional high-quality development, the leader of the development path, and the builder of the basic social security system, plays an important role in promoting the overall progress of regional innovation, coordination, greenness, openness, and sharing.

Fiscal decentralization, as an institutional arrangement to adjust the fiscal autonomy of local governments, is an important factor that affects local governments' fiscal expenditure preferences, improves the public living environment and social wellbeing levels, and ensures the realization of government functions. On the one hand, fiscal decentralization can promote public health and enhance public wellbeing [20]. The Classical Western Fiscal Decentralization Theory suggests that under a fiscal decentralization system, local governments can be motivated to take active measures to ensure the supply of public goods, build healthier lifestyles, and improve the general welfare of society through two mechanisms: "voting with hands" and "voting with feet". The "voting with hands" can directly influence the outcome of local elections, and a government chosen by residents has a relative advantage in terms of information and better knowledge of the preferences of residents in its jurisdiction than the central government. The "voting with feet" enables residents to choose where to live based on their preferences for the level of taxation and the mix of public goods and the goal of utility maximization, ensuring a better match between public goods and residents' preferences and providing incentives for local governments to compete and actively improve the level of provision of local public services [21,22]. Therefore, fiscal

decentralization can effectively improve the quality of regional development by enhancing the supply of public goods and services [23] and optimizing the efficiency of healthcare expenditure [24]. On the other hand, fiscal decentralization can curb local government spending on science and technology [17], distort the government spending structure [25], reduce the level of environmental pollution control [26], and increase regional income inequality [27], which is detrimental to regional quality development and the achievement of the UN Sustainable Development Goals (SDGs).

Uncertain external shocks, represented by public safety and health events such as the COVID-19 pandemic, have prompted economic and social development to focus more on solving livelihood issues based on healthcare and social security employment system improvements, thereby building healthier lifestyles and promoting higher levels of public wellbeing. In this context, the capacity for government collaboration plays an important role [28,29], in which the government affects regional development through actions such as changing spending preferences or spending efficiency. When the share of fiscal spending on health is low, shocks from the COVID-19 pandemic can exacerbate the healthcare crisis [30], while an increase in the share of public fiscal resources spent on healthcare can have a positive impact on economic recovery and high-quality development [31]. At the same time, the quality and level of public demand in the field of healthcare and social security and employment are increasing, which also requires the government to increase fiscal spending on livelihood programs, improve the healthcare system through fiscal means, establish a healthier lifestyle, and thus strengthen the control of the COVID-19 pandemic [32]. However, is there an effect of the moderation of fiscal decentralization on the promotion or suppression of regional development quality? How is the effect on social wellbeing and regional development quality transmitted through changing fiscal expenditure preferences in livelihood areas? How can the fiscal decentralization system be improved in the post-COVID-19 era? This needs to be explored further.

Based on this, this paper explores the impact of fiscal decentralization on regional high-quality development and the transmission mechanisms from the perspective of healthcare, social security employment, and other livelihood expenditures. The marginal contributions of this paper mainly include the following aspects. First, when existing studies explore the relationship between fiscal decentralization and regional economic development, they mostly use a single index, such as regional economic growth or innovation development, to measure the level of economic development. This paper constructs a comprehensive evaluation index of regional high-quality development according to the UN Sustainable Development Goals and the new development concept, which enriches the measurement of the level of high-quality development. Second, most existing studies analyze the impact of fiscal decentralization on regional economic development in terms of innovation and inter-governmental competition. This paper explores the effect of the existing fiscal system on the quality of regional development from the perspective of healthcare, social security employment, and other livelihood expenditures, and effectively explains why China's healthcare infrastructure is strongly supported by the government. Third, this paper puts forward new hypotheses and enriches the findings related to fiscal decentralization and regional high-quality development, arguing that fiscal decentralization has an inverted U-shaped impact on the quality of regional development, revealing that the existing fiscal decentralization system has certain superiorities and limitations according to the livelihood expenditure transmission mechanism, and providing an empirical basis for government fiscal policy formulation and systems improvement in the post-COVID-19 era.

## 2. Theoretical Analysis and Research Hypothesis

High-quality development is the result of the joint action of the government and the market, and government actions have an important impact on regional development. In the post-COVID-19 era, the government needs to promote high-quality regional economic and social development by improving its internal system, and the reform of the decentralization system has an important impact on the government's support for high-quality development

from multiple dimensions. Based on the first generation of fiscal federalism, in the context of fiscal decentralization, local governments can make use of their information advantage to form strong links with the preferences of local residents and optimize the provision of public goods and services under fiscal constraints, thereby maximizing the welfare of local voters and effectively promoting regional development [33–35]. Based on the second generation of fiscal federalism, fiscal decentralization enhances the fiscal autonomy of local governments to promote more prosperous economic activities that generate a larger share of tax revenue through the provision of public goods and services that promote market development, thereby promoting regional economic growth [36]. In the Chinese style of fiscal decentralization, fiscal decentralization under centralized political power effectively enhances the incentive of local governments to promote regional economic growth [37], using the advantages of information to improve institutional arrangements that are compatible with regional development and promote intergovernmental institutional innovation, thereby achieving successful progressive development and reform in the region.

According to theories related to fiscal decentralization, the impact of fiscal decentralization on the quality of regional economic and social development is mainly focused on two aspects when implemented in the context of a high-quality development strategy with the new development concept as its core. On the one hand, from the perspective of the public goods and services supply, local governments have an information advantage in terms of local resource endowments and local residents' demand preferences, and can provide effective supply through fiscal spending to better meet peoples' demands for a better life. With the increase in fiscal decentralization, local governments' fiscal constraints are reduced, and their autonomy is increased, which improves the overall quality of life of the region through the supply of high-quality public goods or services [23,34]. From the perspective of promoting the optimization and upgrading of regional economic structures, fiscal decentralization promotes economic incentives and competition, optimizes industrial structures, and thus achieves structural transformation and economic development [36,38]. As the fiscal autonomy of local governments increases, local governments will adopt fiscal and tax policies such as tax exemptions [39] and environmental regulations to achieve economic restructuring goals and enhance regional economic development under the tax competition mechanism. On the other hand, based on the Economic Human Assumptions (Zhu and Liu, 2021), local governments with information advantages are more inclined to "focus on production rather than innovation" and "focus on production rather than services". With the increase in fiscal decentralization, local governments' development strategy of "competing for growth" is not the same as that of "competing for growth" [40,41]. With the increase in fiscal decentralization, local governments compete for limited capital and high-skilled labor by increasing economic public goods that provide more benefits to capital owners and high-skilled labor, which crowds out spending on non-economic public goods. The development strategy of "competition for growth" by local governments is contrary to the goal of high-quality regional development and thus hinders high-quality regional development. Therefore, based on the above analysis, this paper proposes the following hypothesis.

**Hypothesis 1.** *The impact of fiscal decentralization on regional high-quality development has non-linear characteristics and shows an inverted U-shape.*

In the face of the uncertainties represented by public health and safety events such as the COVID-19 pandemic, a sound healthcare and social security employment system is an important guarantee of economic and social stability and a basic prerequisite for healthy regional development [25,42,43]. As the main builder and maintainer of soft environments such as healthcare and social security employment, local governments support the construction of public areas such as healthcare and social security and employment through fiscal expenditures to ensure the basic quality of the life and health of residents and enhance the resilience of households to risks [44], which maintains the stability of

regional development and indirectly influences the process of high-quality regional development. Fiscal expenditures on healthcare and social security employment promote the redistribution of economic resources and wealth by safeguarding the life and health of residents, stabilizing labor force social security employment and maintaining peoples' basic livelihood security [45], which enhances the degree of regional coordination and sharing and thus contributes to the improvement in regional development quality. In addition, some scholars have used life-cycle theory to explain "The mystery of Savings in China", suggesting that rational individuals have a general precautionary saving behavior [46,47]. Fiscal expenditures on healthcare and social security provide basic social security for residents, resulting in a relative reduction in spending on healthcare and social security, a reduction in precautionary savings, an increase in lifetime real income levels, and an increase in willingness and motivation to consume, which effectively expands the domestic demand market and promotes the construction of an internal circulation development pattern, thereby promoting high-quality regional development.

Under the Chinese decentralization system, the indirect effects of fiscal decentralization on regional development quality will be reflected by changes in local governments' spending preferences for healthcare and social security employment. When local governments that rely on information advantages aim to maximize the public welfare of residents, fiscal decentralization can optimize the efficiency of public resource allocation and expenditure [27], improve the basic environment for human capital development [23], promote human capital accumulation, and thus enhance the level of economic quality. When the government takes economic growth as its main goal, fiscal decentralization may increase fiscal expenditures related to economic growth [25] at the expense of fiscal expenditures that only have the attributes of public goods [41], and there is a profit-seeking motive to show "bottom-up competition" for social fiscal expenditures, attaching less importance to basic livelihood security, which in turn affects the quality level of regional development. From the perspective of government expenditures on peoples' livelihoods, this "bottom-up competition" approach may lead local government expenditure to favor the healthcare sector, where there is a potential economic market, to optimize the efficiency of health care resource use [24] and lead to the establishment of a healthcare consumer market with huge demand. This will lead to the establishment of a strong consumer market for pharmaceuticals and healthcare and promote the high-quality development of the pharmaceutical industry and the regional economy. At the same time, local governments pay less attention to public services with strong social attributes, neglecting the proportion of spending on basic livelihood protection and social security employment assistance, which in turn affects the overall quality of life of residents and the level of regional development. Based on the above analysis, this paper proposes the following hypotheses.

**Hypothesis 2.** *Fiscal decentralization can enhance government healthcare expenditure preferences and promote high-quality regional development.*

**Hypothesis 3.** *Fiscal decentralization can reduce the government's expenditure preference for social security employment and impede high-quality regional development.*

### 3. Research Design
#### 3.1. Regression Model

To verify the effect of fiscal decentralization on the level of regional quality development, Hypothesis 1 is verified. The basic regression model is set up in this paper as follows.

$$hqd_{it} = \beta_0 + \beta_1 fd_{it} + \sum_{j=2}^{7} \beta_j control_{jit} + \varepsilon_{it} \qquad (1)$$

$$hqd_{it} = \beta_0 + \beta_1 fd_{it} + \beta_2 fd_{it}^2 + \sum_{j=3}^{8} \beta_j control_{jit} + \varepsilon_{it} \tag{2}$$

where the *hqd* represents the high-quality development level of the region, the *fd* represents the degree of regional fiscal decentralization, and the *control* is the set of control variables. The *i* denotes provincial-level indicators, the *t* denotes year-level indicators, the $\beta$ is the regression coefficient, and the $\varepsilon$ is a random interference term.

To verify Hypotheses 2 and 3, we explore the transmission path of fiscal decentralization through changing government expenditure preferences on healthcare and social security and employment, thus affecting regional high-quality development from the perspective of livelihood-based expenditure. This paper draws on the design of the mediation model by Baron et al. [48] and uses the stepwise regression method to test the mediation effect, where the *fd_hc* and the *fd_sse* are local government healthcare expenditure intensity and social security employment expenditure intensity, respectively.

$$fd\_hc_{it}|fd\_sse_{it} = \gamma_0 + \gamma_1 fd_{it} + \sum_{j=2}^{7} \gamma_j control_{jit} + \varepsilon_{it} \tag{3}$$

$$hqd_{it} = \varphi_0 + \varphi_1 fd_{it} + \varphi_2 fd\_hc_{it}|fd\_sse_{it} + \sum_{j=3}^{8} \varphi_j control_{jit} + \varepsilon_{it} \tag{4}$$

### 3.2. Variable Description

#### 3.2.1. Dependent Variable

Existing studies have mostly used total factor productivity (TFP) as a measure of quality development, which is not a comprehensive reflection of the connotation of high-quality economic development in the United Nations Sustainable Development Goals (SDGs). In this paper, based on the 2015 UN SDGs and the new development concept proposed by the Chinese government, while adhering to the principles of feasibility and simplicity in the construction of evaluation indicators, as well as ensuring the quality and result orientation of indicator selection [10], we will construct an evaluation system from five dimensions: innovation, coordination, greenness, openness and sharing. The specific indicators are shown in Table 1.

**Table 1.** Index of high-quality development level.

| First-Level Indicators | Second-Level Indicators | Third-Level Indicators | Notes | References |
|---|---|---|---|---|
| High-quality development level (*hqd*) | Innovation ($Z_1$) | Degree of emphasis on scientific and technological innovation ($Z_{11}$) | Full-time equivalent of R&D personnel $Z_{11}$ | [49–51] |
| | | Technology research and development capability ($Z_{12}$) | Number of invention patents authorized $Z_{12}$ | [50,51] |
| | | Technology transformation capability ($Z_{13}$) | Development degree of high-tech industry (Main Business Income/GDP) $Z_{13}$ | [50,51] |
| | Coordination ($Z_2$) | Level of coordinated development of regional industries ($Z_{21}$) | Degree of rationalization of industrial structure $Z_{21}$ | [52,53] |
| | | Level of coordinated urban–rural development ($Z_{22}$) | Urban–rural income gap (Income of Rural Residents/Income of Urban Residents) $Z_{22}$ | [53] |

| First-Level Indicators | Second-Level Indicators | Third-Level Indicators | Notes | References |
|---|---|---|---|---|
| | Greenness ($Z_3$) | Basic environmental change degree ($Z_{31-34}$) | $PM_{2.5}$ population weighted value $Z_{31}$; Unit energy consumption to create GDP value $Z_{32}$; Industrial solid waste utilization rate $Z_{33}$; Urban wastewater utilization rate $Z_{34}$ | [54,55] |
| | | Development of environmental protection technology ($Z_{35}$) | Number of green invention patents granted (mainly including seven categories such as alternative energy, transportation, energy conservation, waste management, agriculture and forestry, administrative supervision and design, and nuclear power) $Z_{35}$ | |
| | Openness ($Z_4$) | External Attractiveness ($Z_{41}$) | Actual amount of foreign direct investment in the region $Z_{41}$ | [56] |
| | | Internal Marketization Level ($Z_{42}$) | Regional Marketization Index $Z_{42}$ | [57] |
| | | Informatization Development Status ($Z_{43}$) | Regional Internet penetration rate $Z_{43}$ | [58] |
| | Sharing ($Z_5$) | Peoples' quality of life ($Z_{51}$) | Consumption level of regional residents $Z_{51}$ | [59] |
| | | Social civilization level ($Z_{52}$) | Number of college students/regional population $Z_{52}$ | |
| | | Life and health security ($Z_{53-54}$) | Number of hospital beds for ten thousand people $Z_{53}$; Number of practicing (assistant) physicians for ten thousand people $Z_{54}$ | |
| | | Basic social security ($Z_{55-57}$) | Labor resource utilization rate $Z_{55}$; Medical insurance coverage ratio $Z_{56}$; Unemployment insurance coverage ratio $Z_{57}$ | |

In the process of indicator construction, this paper draws on the treatment of Ding [60] and uses the entropy-weighted TOPSIS model to calculate the high-quality development index of 30 provinces (cities and districts) from 2006 to 2018 through the objective weighting method. The specific steps are as follows.

First, we standardize each indicator according to Equation (5). The $x_{ij}$ and the $x_{ij}$ denote the original and standardized values of the $j$ indicator of the $i$ evaluation object, respectively. The $x^+{}_{ij}$ (+) and the $x^-{}_{ij}$ (−) are the calculation processes for positive and negative indicators, respectively.

$$x'_{ij}(+) = \frac{x_{ij} - \min\left\{x_{1j}, \cdots x_{ij}\right\}}{\max\left\{x_{1j}, \cdots x_{ij}\right\} - \min\left\{x_{1j}, \cdots x_{ij}\right\}}$$
$$x'_{ij}(-) = \frac{\max\left\{x_{1j}, \cdots x_{ij}\right\} - x_{ij}}{\max\left\{x_{1j}, \cdots x_{ij}\right\} - \min\left\{x_{1j}, \cdots x_{ij}\right\}}$$

$$(5)$$

Second, combining the results of Equation (5), we calculate the information entropy value $e_j$ for each indicator according to Equation (6).

$$e_j = -\frac{1}{\ln n} \sum_{i=1}^{n} \left[ \left( \frac{x'_{ij}}{\sum_{i=1}^{n} x'_{ij}} \right) \ln \left( \frac{x'_{ij}}{\sum_{i=1}^{n} x'_{ij}} \right) \right] \quad (6)$$

Third, combining the results of Equation (6), according to Equations (7) and (8), we further obtain the weight values $w_j$ and the weighting index $z_{ij}$ for each indicator.

$$w_j = \frac{1 - e_j}{\sum_{j=1}^{m} (1 - e_j)} \quad (7)$$

$$z_{ij} = w_j \times x'_{ij} \quad (8)$$

Fourth, according to Equation (9), we calculate the Euclidean distance $D^+_i$, $D^-_i$ of each evaluation object from the positive and negative ideal solutions.

$$D^+_i = \sqrt{\sum_{j=1}^{m} \left( z^+_{ij} - z_{ij} \right)^2}, \ D^-_i = \sqrt{\sum_{j=1}^{m} \left( z_{ij} - z^-_{ij} \right)^2} z^+_{ij} = \max\{z_{1j}, \cdots, z_{ij}\}, z^-_{ij} = \min\{z_{1j}, \cdots, z_{ij}\} \quad (9)$$

Fifth, we calculate the relative closeness of each evaluation object to the ideal solution according to Equation (10), that is, the final evaluation index $C_i$.

$$C_i = \frac{D^-_i}{D^+_i + D^-_i} \quad (10)$$

### 3.2.2. Independent Variable

To explore the Chinese style of fiscal decentralization, appropriate fiscal decentralization indicators should be constructed. Most of the existing fiscal decentralization indicators are measured by the ratio of local fiscal expenditures to central fiscal expenditures. Under centralized political power, fiscal decentralization indicators have the same denominator in the same year and depend mainly on the relative size of fiscal revenues and expenditures of each province, which is not a good measure of the degree of fiscal decentralization in China. Lin and Liu [61] refined the fiscal decentralization indicator by using the marginal share of provincial governments in the budget revenue of the province to measure the degree of decentralization of Chinese provinces with the central government, but the indicator cannot reflect the intertemporal changes in the degree of fiscal decentralization. To better measure the degree of fiscal decentralization in China, we construct fiscal decentralization indicators based on the treatment of Zhu et al. [62,63], as shown in Equation (11).

$$fd\_exp = \frac{\frac{FE_i}{POP_i}}{\frac{FE_i}{POP_i} + \frac{FE_c}{POP_n}} \times \left( 1 - \frac{GDP_i}{GDP_n} \right) \quad (11)$$

where the *fd_exp* denotes the degree of fiscal expenditure decentralization. The *FE* denotes the general budget expenditure of the government at this level. The *GDP* is the gross regional product. The *POP* denotes the regional population size. And *i*, *c*, and *n* denote provincial, central, and national indicators, respectively. The greater the value of *fd_exp* is, the higher the degree of fiscal decentralization.

### 3.2.3. Mediating Variables

In the post-COVID-19 era, the importance and fiscal support given by governments to healthcare and social security and employment among livelihood expenditure items

have increased significantly compared to other fiscal expenditure items. Therefore, we analyze healthcare expenditure preferences (*fd_hc*) and social security and employment expenditure preferences (*fd_sse*) to represent the basic livelihood expenditure preference items, which are represented by the percentages of healthcare fiscal expenditure and social security and employment fiscal expenditure, respectively.

### 3.2.4. Control Variables

The control variables involved in the empirical analysis of this paper mainly include R&D support (*r&d*), innovation foundation and innovation atmosphere (*patent*), level of external openness (*fdi*), *energy* consumption level (*energy*), economic growth rate (*gdp*), and population agglomeration degree (*upd*). Specifically, the *r&d* represents the region's emphasis on scientific and technological development, which is measured by the rate of change in R&D investment. The *patent* represents the regional innovation foundation and atmosphere, which is measured by the number of invention patents granted per 10,000 people. The *fdi* represents the level of opening to the outside world, measured as the actual amount of foreign direct investment per capita. The *energy* represents the importance of green development in the region and is measured by the energy consumption per unit of GDP. The *gdp* represents the rate of regional economic growth, measured as the rate of change in GDP. The *upd* represents the degree of population agglomeration in the region, measured by the urban population density. Finally, the descriptive statistics of the relevant variables are shown in Table 2.

**Table 2.** Descriptive statistics for variables.

| Variables | Sample Size | Mean | Median | Standard Deviation | Min. Value | Max. Value |
|---|---|---|---|---|---|---|
| *hqd* | 390 | 0.379 | 0.372 | 0.108 | 0.166 | 0.706 |
| *innovation* | 390 | 0.162 | 0.108 | 0.159 | 0.005 | 0.958 |
| *coordination* | 390 | 0.519 | 0.520 | 0.166 | 0.000 | 0.911 |
| *greeness* | 390 | 0.449 | 0.448 | 0.092 | 0.249 | 0.815 |
| *openness* | 390 | 0.385 | 0.356 | 0.159 | 0.106 | 0.805 |
| *sharing* | 390 | 0.339 | 0.330 | 0.133 | 0.088 | 0.845 |
| *fd_exp* | 390 | 0.806 | 0.807 | 0.064 | 0.632 | 0.933 |
| *fd_hc* | 390 | 0.068 | 0.067 | 0.017 | 0.026 | 0.106 |
| *fd_sse* | 390 | 0.129 | 0.131 | 0.034 | 0.056 | 0.275 |
| *gdp* | 390 | 0.133 | 0.121 | 0.063 | −0.040 | 0.298 |
| *fdi* | 390 | 0.109 | 0.072 | 0.132 | 0.000 | 0.851 |
| *energy* | 390 | 1.060 | 0.904 | 0.613 | 0.224 | 4.142 |
| *r&d* | 390 | 2.164 | −0.192 | 7.604 | −0.946 | 66.562 |
| *patent* | 390 | 1.223 | 0.449 | 2.459 | 0.036 | 21.810 |
| *upd* | 390 | 0.280 | 0.258 | 0.122 | 0.060 | 0.631 |

### 3.3. Data Sources

On the one hand, the reform of fiscal expenditure items occurred after 2006, so we need to ensure the consistency of the statistical caliber of the data. In addition, the Eleventh Five-Year Plan in 2006 clearly pointed out that "adhering to the people-oriented approach, changing the development concept, innovating the development model, improving the quality of development, implementing the "five coordinations", and effectively turn economic and social development into the track of comprehensive, coordinated and sustainable development". The focus of economic and social development has begun to tilt toward innovation-driven and high-quality development, emphasizing comprehensive and sustainable development. On the other hand, we needed to exclude the impact of the COVID-19 pandemic on economic development since 2019 in the design of our study, and the marketability index, an important indicator of the openness of regional development, is publicly available as of 2018. Therefore, to ensure the integrity and reliability of the data, we use the relevant data for 30 provincial regions (excluding Tibet, Hong Kong, Macao, and Taiwan) from 2006–2018.

The data sources we used are listed as follows. First, we have several sources for the data involved in the construction process of regional quality development indicators. Specifically, the original data on the degree of emphasis on scientific and technological innovation ($Z_{11}$) and technology research and development capabilities ($Z_{12}$) are obtained from the China Science and Technology Statistical Yearbook. The original data on technology transformation capabilities ($Z_{13}$) are obtained from the China High Technology Industry Statistical Yearbook and the database of the National Bureau of Statistics. The level of coordinated development of regional industries ($Z_{21}$) is measured by the Thiel index. This paper uses the Thiel index to measure the degree of rationalization of industrial structures, and the calculation formula is $z_{i,t} = \sum_{m=1}^{3} y_{i,m,t} \cdot \ln(y_{i,m,t}/l_{i,m,t}), m = 1, 2, 3$. The $y_{i,m,t}$ represents the proportion of the $m$ industry in region $i$ in the GDP in period $t$, and the $l_{i,m,t}$ represents the proportion of the $m$ industry in region $i$ in the total employment from one person in period $t$; the smaller the value of $z$, the more reasonable the industrial structure. The original data of PM$_{2.5}$ are obtained from the global raster data of Washington University in St. Louis, USA, https://sites.wustl.edu/acag/datasets (accessed on 8 August 2022). The number of green patents granted in the region ($Z_{35}$) is based on the number of green invention patents granted to listed companies in the patent database of the China National Intellectual Property Administration, https://www.cnipa.gov.cn/col/col1510/index.html (accessed on 8 August 2022). The marketization level index ($Z_{42}$) is obtained from the China Market Index Database, https://cmi.ssap.com.cn/dataQuery (accessed on 8 August 2022). The original data on the remaining indicators are obtained from the database of the Chinese National Bureau of Statistics, and among the three-level indicators, $Z_{21}$ and $Z_{31}$ are negative indicators, while the rest are positive indicators. In addition, the original data of the independent variable, control variables, and mediating variables are obtained from the China Statistical Yearbook and the database of the Chinese National Bureau of Statistics.

## 4. Regression Analysis

### 4.1. Evaluation and Analysis of Regional High-Quality Development Indexes

Table 3 shows the average value and rankings of the comprehensive index of high-quality development for each region from 2006 to 2018. Figure 1 shows the regional visualization results of the level of high-quality development. In terms of the provincial comprehensive index of high-quality development, the highest average value of the index is 0.582 in Beijing, followed by Guangdong, Shanghai, and Jiangsu, which are all above 0.5 and in the first echelon. Some provinces, including Zhejiang, Tianjin, Shandong, and Fujian, have a composite index average value of 0.4–0.5, which is in the second echelon. The remaining 22 provinces have a comprehensive index average value below 0.4, of which 19 provinces are in the third echelon with a small difference of 0.3–0.4, with more room for improvement. In addition, some provinces, such as Qinghai, Guizhou, and Gansu, have a comprehensive development index below 0.3 and a low level of high-quality development. This reflects that the level of high-quality development in China presents more obvious ladder-type characteristics, but also reflects the regional imbalance of high-quality development; the overall level of development is not high.

**Table 3.** The Regional High-Quality Development Index.

| Region | *hqd* | Sort | Region | *hqd* | Sort |
|---|---|---|---|---|---|
| Beijing | 0.582 | 1 | Henan | 0.359 | 15 |
| Tianjin | 0.482 | 6 | Hubei | 0.367 | 13 |
| Hebei | 0.326 | 23 | Hunan | 0.342 | 18 |
| Shanxi | 0.330 | 22 | Guangdong | 0.540 | 2 |
| Inner Mongolia | 0.337 | 19 | Guangxi | 0.302 | 26 |
| Heilongjiang | 0.360 | 14 | Hainan | 0.375 | 11 |
| Jilin | 0.368 | 12 | Chongqing | 0.386 | 10 |
| Liaoning | 0.392 | 9 | Sichuan | 0.330 | 21 |
| Shanghai | 0.531 | 3 | Guizhou | 0.285 | 29 |
| Jiangsu | 0.516 | 4 | Yunnan | 0.301 | 27 |
| Zhejiang | 0.491 | 5 | Shaanxi | 0.333 | 20 |
| Anhui | 0.354 | 16 | Gansu | 0.282 | 30 |
| Fujian | 0.412 | 8 | Ningxia | 0.312 | 25 |
| Jiangxi | 0.348 | 17 | Qinghai | 0.288 | 28 |
| Shandong | 0.422 | 7 | Xinjiang | 0.318 | 24 |

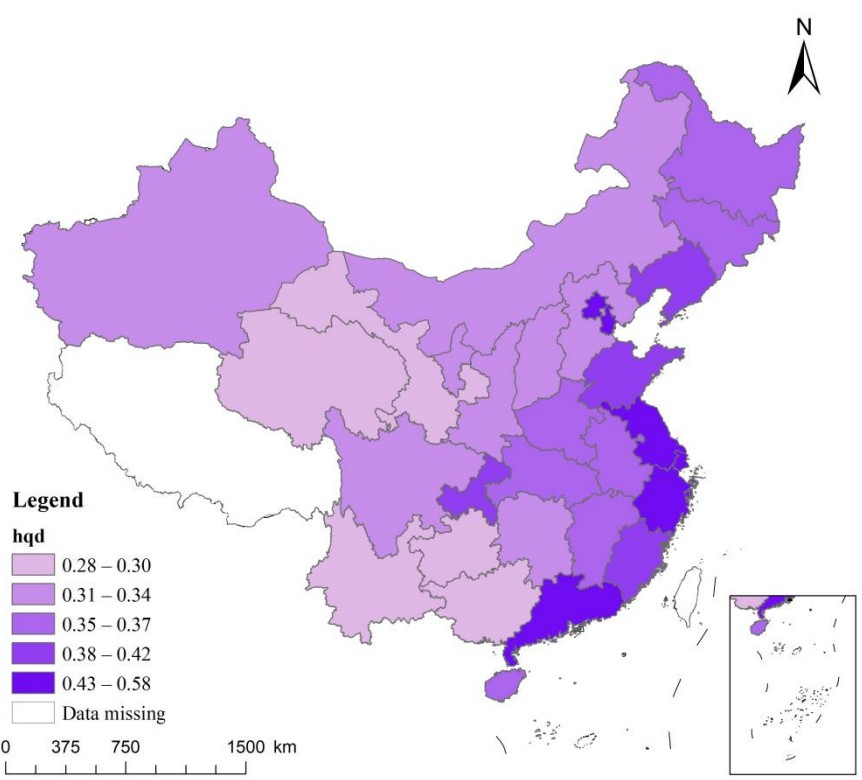

**Figure 1.** Average Ranking of Sustainable Development of Chinese provinces (Drawing Review No. GS(2019)1822), http://bzdt.ch.mnr.gov.cn/download.html?searchText=GS(2019)1822 (accessed on 8 August 2022).

### 4.2. Baseline Regression Analysis

To test the effect of fiscal decentralization on regional development quality, this paper conducts a stepwise regression of Equations (1) and (2) using the Tobit random effects model, random effects model, mixed effects model, individual fixed effects model, and two-way fixed effects model from model 1 to model 5, respectively, and model 6 is the regression result of further adding the quadratic term of fiscal decentralization, as shown in Table 4. From model 1 to model 5, the regression coefficient of fiscal decentralization on regional development quality is always positive and passes the 5% confidence level test, indicating that fiscal decentralization can promote regional development quality. After adding the quadratic term of fiscal decentralization, the primary and quadratic

terms of fiscal decentralization are one positive and one negative, respectively. Both of them passed the confidence level test below 1%, which reflects that the impact of fiscal decentralization on regional development quality shows an inverted U-shape, and the effect of fiscal decentralization is moderate. The main reason may be that fiscal decentralization changes local government fiscal spending preferences, which indirectly affects the level of regional development quality, thus showing an inverted U-shape in the overall regression.

**Table 4.** Baseline regression result.

| | (1) hqd | (2) hqd | (3) hqd | (4) hqd | (3) hqd | (4) hqd |
|---|---|---|---|---|---|---|
| *fd_exp* | 0.580 *** | 0.540 *** | 0.221 ** | 0.636 *** | 0.525 *** | 1.847 *** |
| | (0.061) | (0.056) | (0.100) | (0.119) | (0.073) | (0.422) |
| *fd_exp2* | | | | | | −0.908 *** |
| | | | | | | (0.286) |
| *gdp* | −0.203 *** | −0.211 *** | −0.367 *** | −0.196 *** | −0.021 | −0.017 |
| | (0.029) | (0.030) | (0.046) | (0.027) | (0.035) | (0.035) |
| *patent* | 0.017 *** | 0.016 *** | 0.013 ** | 0.018 *** | 0.005 *** | 0.005 *** |
| | (0.001) | (0.001) | (0.006) | (0.005) | (0.001) | (0.001) |
| *energy* | −0.085 *** | −0.088 *** | −0.076 *** | −0.080 *** | 0.001 | −0.005 |
| | (0.007) | (0.007) | (0.012) | (0.017) | (0.007) | (0.007) |
| *r&d* | 0.002 *** | 0.002 *** | −0.000 | 0.003 ** | 0.000 | 0.000 |
| | (0.001) | (0.001) | (0.001) | (0.002) | (0.000) | (0.000) |
| *fdi* | 0.002 | 0.020 | 0.167 *** | −0.020 | 0.021 | 0.029 |
| | (0.030) | (0.029) | (0.050) | (0.041) | (0.019) | (0.019) |
| *upd* | 0.042 | 0.028 | −0.096 ** | 0.051 | 0.011 | 0.004 |
| | (0.029) | (0.029) | (0.043) | (0.060) | (0.019) | (0.019) |
| *constant* | −0.009 | 0.032 | 0.323 *** | −0.065 | −0.055 | −0.520 *** |
| | (0.056) | (0.051) | (0.097) | (0.114) | (0.059) | (0.158) |
| Individual fixed | | | | YES | YES | YES |
| Time fixed | | | | | YES | YES |
| Wald test | 2049.03 *** | 1908.45 *** | | | | |
| adj. $R^2$ | | | 0.785 | 0.848 | 0.973 | 0.974 |
| *N* | 390 | 390 | 390 | 390 | 390 | 390 |

Notes: The values in brackets are standard deviations. *** and ** indicate that the estimated coefficients are significant at the confidence levels of 1% and 5%, respectively.

From the regression results of the control variables, the regression coefficients of economic growth speed for regional high-quality development are all negative and pass the 1% significance test in models 1 to 4, indicating that the past crude economic development model of simply pursuing growth speed is contrary to the direction of high-quality development with innovation, coordination, greenness, openness, and sharing as the core. Therefore, it is not desirable to blindly pursue economic growth speed in the period of economic transition, and more attention should be given to the issue of economic development quality. The regression coefficients of innovation base and atmosphere for regional high-quality development are all positive, and all pass the significance test at a confidence level of 5% or more, which effectively verifies the importance of building an innovation environment for regional high-quality development. The level of energy consumption has a significant negative impact on regional high-quality development and passes the 1% significance test in models 1 to 4, which means that the reduction in the energy consumption level per unit of GDP can promote regional green development and is conducive to enhancing regional high-quality development. The regression coefficients of R&D support for the level of regional high-quality development are positive and pass significance tests above 5% in models 1, 2, and 4, indicating that the investment in valuing innovation resources will effectively improve the quality of regional economic development, mainly because increasing financial investment promotes the concentration of talent and provides human capital for regional innovation-driven development. The regression coefficient of

the level of foreign openness for regional high-quality development is positive and passes the 1% significance test in model 3, which indicates that there is a positive spillover effect of the level of foreign investment on regional high-quality development, and effectively attracting foreign investment is an important initiative for economic development. Most of the regression coefficients of the population concentration degree are positive, but they do not pass the significance test. We should further optimize the regional population spatial structure and construct a benign relationship between urban population density and regional high-quality development.

To test the robustness of the effect of fiscal decentralization on regional high-quality development, this paper conducts robustness tests by replacing the analytical model, lagging behind the core explanatory variables and regression analysis on specific dimensions of high-quality development. The results are shown in Table 5. Model 1 is the regression result of the Tobit model, and the inverted U-shaped relationship between fiscal decentralization and regional high-quality development is significant at the 1% confidence level. Models 2 to 6 are the results of regressions with innovation, coordination, greenness, openness, and shared development indices as explanatory variables, respectively. Among them, the inverted U-shaped relationship between fiscal decentralization and regional innovation, greenness, and openness development levels passes the 5% confidence level test or above, which indicates that the promotion effect of increased fiscal decentralization on regional development is moderate in multiple dimensions.

**Table 5.** The robustness checks.

| | (1) hqd | (2) Innovation | (3) Coordination | (4) Greenness | (5) Openness | (6) Sharing |
|---|---|---|---|---|---|---|
| fd_exp | 2.322 *** | 7.271 *** | 0.720 | 2.827 *** | 1.267 ** | −0.093 |
| | (0.634) | (0.991) | (0.852) | (0.667) | (0.570) | (0.618) |
| fd_exp2 | −1.145 *** | −4.276 *** | −0.076 | −1.432 *** | −0.963 ** | 0.562 |
| | (0.414) | (0.671) | (0.577) | (0.451) | (0.386) | (0.418) |
| constant | −0.660 *** | −2.955 *** | −0.045 | −0.895 *** | −0.052 | 0.074 |
| | (0.242) | (0.370) | (0.318) | (0.249) | (0.213) | (0.231) |
| control | YES | YES | YES | YES | YES | YES |
| Individual fixed | | YES | YES | YES | YES | YES |
| Time fixed | | YES | YES | YES | YES | YES |
| Wald test | 2090.45 *** | | | | | |
| adj. $R^2$ | | 0.934 | 0.956 | 0.912 | 0.978 | 0.964 |
| N | 390 | 390 | 390 | 390 | 390 | 390 |

Notes: The values in brackets are standard deviations. *** and ** indicate that the estimated coefficients are significant at the confidence levels of 1% and 5%, respectively.

This paper eliminates the endogeneity problem through the instrumental variables method, using the lagged one-period fiscal decentralization (*L.fd_exp*) as the instrumental variable. Specifically, we use the IV-two-stage least squares regression (IV−2SLS) method, optimal generalized methods of moments (GMM), and the iterative GMM to alleviate the potential endogeneity issue. The analysis results are shown in Table 6. The *p*-value of the F test is 0.000, and the Minimum Eigenvalue Statistic of 21.157 is greater than 10, which indicates that there is no problem with weak instrumental variables. The two-stage least squares (2SLS), optimal GMM, and iterative GMM estimation results from model 1 to model 3 show that the regression coefficients of the primary and quadratic terms of fiscal decentralization are still significant at the 1% confidence level, and the results of optimal GMM estimation and iterative GMM estimation are basically consistent with those of IV-2SLS. The inverted U-shaped relationship between fiscal decentralization and regional high-quality development was further verified. In addition, to eliminate possible two-way causality in the regression model, this paper uses the one-period lagged degree of fiscal decentralization as the explanatory variable, and the regression results are shown in Model 4, where the inverted U-shaped relationship between fiscal decentralization and regional

development quality is significant at the 1% confidence level. In summary, the inverted U-shaped relationship between fiscal decentralization and regional development quality is robust, and there is a moderate promotion effect of fiscal decentralization on regional high-quality development. Hypothesis 1 is verified.

**Table 6.** Endogeneity test.

| | (1)<br>*hqd* | (2)<br>*hqd* | (3)<br>*hqd* | (4)<br>*hqd* |
|---|---|---|---|---|
| *fd_exp* | 31.332 *** | 31.332 *** | 31.332 *** | |
| | (8.097) | (8.097) | (8.097) | |
| *fd_exp2* | −19.429 *** | −19.429 *** | −19.429 *** | |
| | (5.049) | (5.049) | (5.049) | |
| *L.fd_exp* | | | | 1.783 *** |
| | | | | (0.411) |
| *L.fd_exp2* | | | | −0.892 *** |
| | | | | (0.278) |
| *constant* | −12.152 *** | −12.152 *** | −12.152 *** | −0.481 *** |
| | (3.254) | (3.254) | (3.254) | (0.155) |
| *control* | YES | YES | YES | YES |
| *Shea's adj. partial R²* | 0.0381 | | | |
| *Robust F* | 16.7002 | | | |
| | [0.000] | | | |
| *Minimum eigenvalue statistic* | 21.157 | | | |
| adj. *R²* | 0.301 | 0.301 | 0.301 | 0.974 |
| N | 390 | 390 | 390 | 360 |

Notes: The values in brackets "( )" are standard deviations. The values in brackets "[ ]" are the *p*-values of the corresponding test statistics. The values in brackets are standard deviations. *** indicate that the estimated coefficients are significant at the confidence levels of 1%.

*4.3. Analysis of the Transmission Mechanism*

To test hypothesis 2, this paper uses a mediating effects model to regress Equations (3) and (4) from the perspective of healthcare and social security employment expenditure, and combines the regression results of Equation (1) to explore the transmission mechanism of fiscal decentralization on regional development quality to explain the reasons for the inverted U-shaped relationship between fiscal decentralization and regional development quality. The results are shown in Table 7. The regression coefficient of fiscal decentralization on the level of regional high-quality development is significantly positive and passes the 1% confidence level test. The results of the mediating effect test on healthcare expenditure preferences show that the regression coefficient of fiscal decentralization for government healthcare expenditure preference is significantly positive at the 1% confidence level, and the effects of both fiscal decentralization and health care expenditure preference on regional high-quality development are significantly positive at the 1% confidence level, which indicates that fiscal decentralization will enhance the government's role in soft environment construction by increasing the proportion of government healthcare expenditure, and thus promote regional high-quality development. The mediating effect is 0.247 (0.163 × 1.517). The results of the mediating effect of social security employment expenditure preferences show that the regression coefficient of fiscal decentralization on the government's social security employment expenditure preference is significantly negative at the 1% confidence level, and the effects of both fiscal decentralization and social security employment expenditure preference on regional high-quality development are significantly positive at the 1% confidence level. This indicates that as the degree of fiscal decentralization increases, local governments pay relatively less attention to the quality-of-life protection of regional residents, which makes the intensity of social security employment expenditures decrease, which in turn affects the level of regional development quality. The specific masking effect is −0.094 (−0.253 × 0.371). The mediating mechanism was tested by the Sobel estimation

method and the Bootstrap sampling method. The mediating effect of health expenditure preferences was 0.248, the masking effect of social security employment expenditure preferences was 0.094, and the mediating effect of health expenditure preferences and the masking effect of social security employment expenditure preferences existed.

**Table 7.** The regression result of the mediating effect based on healthcare and social security employment expenditure preferences.

|  | *hqd* | *fd_hc* | *hqd* | *fd_sse* | *hqd* |
|---|---|---|---|---|---|
| *fd_exp* | 0.525 *** | 0.163 *** | 0.388 *** | −0.253 *** | 0.730 *** |
|  | (0.073) | (0.030) | (0.120) | (0.048) | (0.150) |
| *fd_hc* | 0.525 *** |  | 1.517 *** |  |  |
|  | (0.073) |  | (0.312) |  |  |
| *fd_sse* |  |  |  |  | 0.371 * |
|  |  |  |  |  | (0.203) |
| *constant* | −0.055 | −0.046 | 0.004 | 0.370 *** | −0.203 |
|  | (0.059) | (0.027) | (0.105) | (0.043) | (0.161) |
| *control* | YES | YES | YES | YES | YES |
| Individual fixed | YES | YES | YES | YES | YES |
| Time fixed | YES | YES | YES | YES | YES |
| adj. $R^2$ | 0.973 | 0.792 | 0.940 | 0.788 | 0.932 |
| *N* | 390 | 390 | 390 | 390 | 390 |
| *Sobel test* |  | 0.248 *** |  | −0.094 *** |  |
|  |  | (0.041) |  | (0.028) |  |
| *Bootstrap test* |  | 0.248 *** |  | −0.094 * |  |
|  |  | (0.046) |  | (0.048) |  |

Notes: The values in brackets are standard deviations. *** and * indicate that the estimated coefficients are significant at the confidence levels of 1% and 10%, respectively.

Each subdimension of high-quality development, on the one hand, is shown in Table 8. Improving healthcare expenditure preferences can promote the innovation, coordination, greenness, openness, and sharing levels of regional development, and all of them pass the significance level test below 5%. The results of the Sobel test and the Bootstrap test support the existence of a mediating effect at the 1% confidence level. Among them, the effect of healthcare expenditure preferences as a mediating effect on the innovation and sharing of regional development is more obvious, with mediating effects of 0.354 and 0.315, respectively. The mediating effects on the coordination, openness, and green development of regional development are 0.234, 0.227, and 0.187, respectively, which indicates that with the increase in fiscal autonomy, local governments can have an all-around effect on regional development quality by enhancing healthcare expenditure preferences, with a more pronounced effect on regional innovation development and shared development levels.

On the other hand, as shown in Table 9, the regression coefficients of social security employment expenditure preferences on the innovation, coordination, greenness, openness, and sharing levels of regional development are all positive, but only the regression coefficients on the innovativeness and sharing of regional development pass the significance test below 10%, which indicates that the transmission mechanism of the government's social security employment underwriting effect on the quality improvement of regional development is not perfect, while it has a more obvious effect on the innovative vitality and sharing of regional development. The regression coefficient of the fiscal decentralization degree on social security employment expenditure preferences is significantly negative at the 1% confidence level. Thus, fiscal decentralization slows down the process of regional high-quality development by reducing the preference for social security employment expenditure. The Sobel test and the Bootstrap test show that the mediating effects of social security employment expenditure preferences on regional development coordination and sharing are more obvious, −0.152 and −0.149, respectively. The mediating effects on regional innovative development and open development are −0.068 and −0.081, respectively.

The mediating effects on regional green development are not significant. This indicates that there are differences in the masking effects of social security employment expenditure preferences on region-specific development indicators. In summary, *hypotheses 2* and *3* were tested.

**Table 8.** Mediating effects of healthcare expenditure preferences based on segmentation dimensions.

| | (1) fd_hc | (2) hqd | (3) Innovation | (4) Coordination | (5) Greeness | (6) Openness | (7) Sharing |
|---|---|---|---|---|---|---|---|
| fd_exp | 0.163 *** | 0.388 *** | 0.402 ** | 0.264 | 0.546 *** | 0.318 * | 0.473 ** |
| | (0.030) | (0.120) | (0.158) | (0.207) | (0.143) | (0.176) | (0.204) |
| fd_hc | | 1.517 *** | 2.166 *** | 1.431 ** | 1.141 *** | 1.388 *** | 1.929 *** |
| | | (0.312) | (0.542) | (0.622) | (0.313) | (0.496) | (0.396) |
| constant | −0.046 | 0.004 | −0.397 ** | 0.258 | −0.054 | 0.058 | −0.065 |
| | (0.027) | (0.105) | (0.161) | (0.200) | (0.136) | (0.133) | (0.165) |
| control | YES | YES | YES | YES | YES | YES | YES |
| Individual fixed | YES | YES | YES | YES | YES | YES | YES |
| Time fixed | YES | YES | YES | YES | YES | YES | YES |
| adj. $R^2$ | 0.792 | 0.940 | 0.931 | 0.924 | 0.904 | 0.938 | 0.901 |
| N | 390 | 390 | 390 | 390 | 390 | 390 | 390 |
| Sobel test | | 0.248 *** | 0.354 *** | 0.234 *** | 0.187 *** | 0.227 *** | 0.315 *** |
| | | (0.041) | (0.062) | (0.059) | (0.039) | (0.052) | (0.060) |
| Bootstrap test | | 0.248 *** | 0.354 *** | 0.234 *** | 0.187 *** | 0.227 *** | 0.315 *** |
| | | (0.047) | (0.051) | (0.051) | (0.053) | (0.013) | (0.069) |

Notes: The values in brackets are standard deviations. ***, **, and * indicate that the estimated coefficients are significant at the confidence levels 1%, 5%, and 10%, respectively.

**Table 9.** Mediating effects of social security employment expenditure preferences based on segmentation dimensions.

| | (1) fd_sse | (2) hqd | (3) Innovation | (4) Coordination | (5) Greeness | (6) Openness | (7) Sharing |
|---|---|---|---|---|---|---|---|
| fd_exp | −0.253 *** | 0.730 *** | 0.824 *** | 0.650 *** | 0.772 *** | 0.625 *** | 0.937 *** |
| | (0.048) | (0.150) | (0.216) | (0.230) | (0.175) | (0.187) | (0.226) |
| fd_sse | | 0.371 * | 0.267 | 0.600 ** | 0.156 | 0.318 | 0.587 * |
| | | (0.203) | (0.163) | (0.243) | (0.135) | (0.254) | (0.324) |
| constant | 0.370 *** | −0.203 | −0.595 *** | −0.029 | −0.164 | −0.123 | −0.370 |
| | (0.043) | (0.161) | (0.198) | (0.250) | (0.166) | (0.191) | (0.240) |
| control | YES | YES | YES | YES | YES | YES | YES |
| Individual fixed | YES | YES | YES | YES | YES | YES | YES |
| Time fixed | YES | YES | YES | YES | YES | YES | YES |
| adj. $R^2$ | 0.788 | 0.932 | 0.921 | 0.923 | 0.896 | 0.934 | 0.894 |
| N | 390 | 390 | 390 | 390 | 390 | 390 | 390 |
| Sobel test | | −0.094 *** | −0.068 * | −0.152 *** | −0.039 | −0.081 ** | −0.149 *** |
| | | (0.028) | (0.040) | (0.046) | (0.026) | (0.037) | (0.043) |
| Bootstrap test | | −0.094 * | −0.068 * | −0.152 *** | −0.039 | −0.081 * | −0.149 ** |
| | | (0.048) | (0.038) | (0.053) | (0.034) | (0.046) | (0.061) |

Notes: The values in brackets are standard deviations. ***, **, and * indicate that the estimated coefficients are significant at the confidence levels of 1%, 5%, and 10%, respectively.

## 5. Conclusions and Policy Implications

In the post-epidemic era, countries have paid significantly more attention to the construction of health care and other livelihood protection systems. The reform of the fiscal and taxation system is of great value in building a perfect livelihood protection system and thus promoting the quality of regional development. The public health system established under the support of China's fiscal system has withstood the test of the COVID-19 pandemic. Therefore, it is appropriate to take China as the research object to explore the relationship between fiscal decentralization, livelihood expenditure preferences, and

regional development quality, and the findings of this paper can also provide an empirical basis for the reform of fiscal systems and public policy design in the post-COVID-19 era in various countries. Specifically, this paper theoretically analyzes and empirically tests the effects and causes of the inverted U-shaped effect of fiscal decentralization on regional high-quality development from the perspective of health care, social security employment, and other livelihood expenditures based on the construction of a regional high-quality development index. Based on the results, we obtained several important conclusions and policy implications.

First, we found that the level of high-quality development in each region shows a more obvious ladder-type feature, and there are problems such as overall poor development quality and regional development imbalances. Therefore, local governments should pay attention to the coordination and synchronization of regional development, address the imbalance of regional high-quality development from the dimensions of innovation, coordination, green, openness, and sharing, and promote the construction of a new development patterns, so as to improve the welfare of society and effectively promote regional high-quality development.

Second, we found that there is a significant nonlinear relationship between fiscal decentralization and regional development quality, showing an inverted U-shaped relationship. Further study found that fiscal decentralization can enhance the expenditure preference of health care, improve the construction of health care infrastructure and enhance the capacity of medical security, establish a healthier lifestyle, and enhance the level of public social welfare, thus indirectly promoting regional high-quality development. Fiscal decentralization can weaken the expenditure preference of social security employment and reduce the importance of basic security for residents, thus hindering the overall regional high-quality development. Therefore, in the process of improving the fiscal decentralization system, while enhancing the fiscal autonomy of local governments, local governments should moderately adjust their fiscal expenditure preferences. On the one hand, we should pay attention to the mediating effect of health care spending preferences on the impact of fiscal decentralization on regional high-quality development, and further enhance the government's guidance and support in the field of medicine and health, so as to promote the construction of a regional soft infrastructure environment and continuously optimize the indicators of high-quality development. On the other hand, we should always pay attention to the solution of the basic social security and social security employment problems, gradually improve the basic quality of life of local residents, and create a healthy and stable social development environment, so as to effectively promote the process of regional high-quality development.

In conclusion, in the process of reforming the fiscal decentralization system in the post-COVID-19 era, we should recognize the appropriateness of the distribution of financial and ministerial powers between the central government and local governments, i.e., while enhancing the welfare of the health care system and economic welfare brought about by the local government's power of income distribution, we must see the neglect of employment and basic social security by local governments. In countries with a centralized system such as China, the central government should reshape the incentives and constraint mechanisms of local governments by reforming the fiscal decentralization system, and while decentralizing fiscal power, it should impose directional constraints on material power to build a perfect social welfare system and thus promote high-quality regional development.

**Author Contributions:** Conceptualization, D.W.; methodology, D.W. and E.Z.; software, D.W.; formal analysis, D.W.; resources, H.L.; data curation, E.Z.; writing—original draft preparation, D.W.; writing— review and editing, D.W., E.Z. and H.L.; supervision, H.L.; project administration, H.L.; funding acquisition, H.L. All authors have read and agreed to the published version of the manuscript.

**Funding:** This research was funded by the National Social Science Funding (Project Name: Research on Transformation of State-Owned Assets Supervision from 'Managing Assets' to 'Managing Capital'; The Funding Number: 17BJY164).

**Data Availability Statement:** Not applicable.

**Acknowledgments:** We would like to express our gratitude to all those who helped us during the writing of this article.

**Conflicts of Interest:** The authors declare no conflict of interest.

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
