# Peer review of "Does Fiscal Decentralization Affect Regional High-Quality Development by Changing Peoples’ Livelihood Expenditure Preferences: Provincial Evidence from China"

_land, doi:10.3390/land11091407_

Round 1

Reviewer 1 Report

1. The format of the manuscript is messy. The font size of some words is much bigger than other words, such as Lines 24, 25, 29, 30, 33, 34, 35 and so on, and the ‘correspondence’ in Line 11 appears twice. The typeface in Page 7-8 is different. Please revise the format of the manuscript to be united, and improve the presentation of the manuscript.

2. The abstract is not well-organized. There lacks the methodology of the paper, and the conclusions part is too long. Please reorganize the abstract.

3. The format of references is not the same as Land. Please revise the style of reference.

4. How to choose the indicators in Table 1? What are the criteria? There is no citing of the references or any criteria of choosing the indicators. The number of indicators selected will affect the results, so how to reduce the error generated by this?

5. Why choosing the data from the period of 2006 to 2018? There lacks explanation of choosing the year 2018.

6. The abbreviation in the first line in Table 2 should be used full name.

7. Is this research related to COVID-19? If the data of 2006-2018 are used, it seems no relation with COVID-19. COVID-19 broke out in the end of 2019. Thus, can this study provide any contribution or references to the post-COVID era?

Author Response

Comments :

1."The format of the manuscript is messy. The font size of some words is much bigger than other words, such as Lines 24, 25, 29, 30, 33, 34, 35 and so on, and the ‘correspondence’ in Line 11 appears twice. The typeface in Page 7-8 is different. Please revise the format of the manuscript to be united, and improve the presentation of the manuscript."

Thank you very much for your comments. There was a formatting issue with the manuscript we submitted during the layout of the paper. We have adjusted it according to the editorial layout format. As shown in the revised manuscript.

2."The abstract is not well-organized. There lacks the methodology of the paper, and the conclusions part is too long. Please reorganize the abstract."

Thank you very much for your comments. We have revised the abstract as well as the conclusion as shown on p.1 and p.18-19.

3."The format of references is not the same as Land. Please revise the style of reference."

Thank you for your comments and reminders. We have modified the format of the references according to LAND's format. As shown on p.20-23. 

4."How to choose the indicators in Table 1? What are the criteria? There is no citing of the references or any criteria of choosing the indicators. The number of indicators selected will affect the results, so how to reduce the error generated by this?"

Thank you for your comments. We explain the rationale for the selection of indicators and the superiority of the methodological choice, and we have added relevant contents in the manuscript as shown p.7-8. And the details are as follows.

First, after the United Nations Sustainable Development Goals (SDGs) were proposed in 2015, the governments have successively put forward SDGs in line with their own development aspirations according to the stage of economic development. The Chinese government has put forward the goal of high-quality development on this basis, and has given an official definition of high-quality development: "High-quality development is development that can well meet the people's growing needs for a better life, development that reflects the new development concept, development in which innovation becomes the first driving force, coordination becomes an endogenous feature, greenness becomes the universal form, openness becomes the necessary path, and sharing becomes the fundamental purpose. " Therefore, there is a policy basis for constructing high-quality development indicators from the five dimensions of innovation, coordination, greenness, openness and sharing.

From the perspective of innovation development level, the process of innovation mainly includes multiple stages such as innovation input, innovation output and economic value of innovation results (Wang, 2021), therefore, we refer to the measurement of Wang et al. (2021), Liu (2020) and Wei and Li (2018), and construct a regional innovation development index in terms of Degree of emphasis on scientific and technological innovation, technology research and development capability and technology transformation capability to construct a regional innovation development index.

From the perspective of coordination development level, coordination development of regions includes two aspects of coordination development of industries and coordination development of regions (Wang and Chen, 2022), we further optimize the index based on the construction of coordination development index by Wang and Chen (2022), and adopt degree of rationalization of industrial structure and urban-rural income gap (Zhao and Chang, 2020).

From the perspective of green development level, the evaluation of regional green development should include Basic environmental change degree as well as development of environmental protection technology. therefore, we draw on Sun and Guo (2021), Zhao (2021) and Zhao et al. (2022) to measure the degree of basic environmental change and the number of green innovation patents to measure the development of environmental protection technology.

In terms of openness, we measure it more comprehensively in terms of the ability to attract foreign investment (Zhou et al., 2022), the level of regional marketization (Hua et al., 2021), and the Internet penetration rate (Chen et al., 2022).

In terms of shared development, shared development is reflected in an overall increase in social welfare levels, which we measure using public consumption levels (Dai et al., 2022) and basic social security indicators such as educational attainment, healthcare, and social security and employment, thus reflecting the shared nature of economic development.

Second, we use TOPSIS to measure the quality development index, which can effectively reduce the measurement error by objectively weighting the index to avoid the influence of subjective factors of principal component analysis (PCA) and other measurements.

Third, the evaluation analysis of the results of the regional development quality index in the manuscript shows that the regional variability of our constructed quality development index is basically consistent with the degree of regional economic development, which indicates the reliability of our constructed index.

References:

  1. Analysis of the High-Quality Development of Urban Agglomerations along the Yellow River Basin. J. Environ. Res. Public Health2022, 19, 2484.
  2. Chuanmin Zhao, Rui Xie, Chunbo Ma, Feng Han. Understanding the haze pollution effects of China's development zone program,Energy Economics, 2022, 111, 106078.
  3. Dai, F. ,  Liu, H. ,  Zhang, X. , &  Li, Q. . Does the equalization of public services effect regional disparities in the ratio of investment to consumption? evidence from provincial level in china:.SAGE Open, 2022,12, 265-277.
  4. Dingqing Wang. Research on innovation efficiency of pharmaceutical manufacturing industry based on the perspective of industrial innovation chain. Jilin University, 2021,6.
  5. Hua, X.; Lv, H.; Jin, X. Research on High-Quality Development Efficiency and Total Factor Productivity of Regional Economies in China. Sustainability2021, 13, 8287.
  6. Liu, H.D.; Liu, T. Research on Coupling Coordination Degree between Innovation-driven and High-quality Economic Development. Technol. Prog. Policy2020, 37, 64–71.
  7. Ming Chen;Hongbo Wang. Import technology sophistication and high-quality economic development: evidence from city-level data of China, Economic Research-Ekonomska Istraživanja, 2022, 35:1, 1106-1141. 
  8. Sun, P.L.; Guo, Z.H. Analysis of spatial difference and influencing factors of high-quality economic development. Decis.2021, 37, 123–125.
  9. Wang, X. , Liu, Y. , & Chen, L. . Innovation efficiency evaluation based on a two-stage dea model with shared-input: a case of patent-intensive industry in china. IEEE Transactions on Engineering Management, 2021, 99, 1-15.
  10. Wei, M.; Li, S.H. Construction and Measurement of China’s Economic Growth Quality Evaluation System under the New Normal.Economist 2018, 4, 19–26.
  11. Zhao, R.Y.; Chang, Z.L. The spatial difference of high-quality economic development and the identification of influencing factors. Financ. Econ. Issues 2020, 10, 22–29.
  12. Zhou, Z.B.; Deng, L.; Xiao, H.L.; Wu, S.J.; Liu, W.B. The Impact of Foreign Direct Investment on China’s High-quality EconomicDevelopment---Index DEA and Panel Partition Regression Analysis. J. Manag. Sci. 2022, 13, 11911.

5."Why choosing the data from the period of 2006 to 2018? There lacks explanation of choosing the year 2018."

Thank you for your comments. On the one hand, the COVID-19 pandemic is an outbreak in 2019, and our study data as of 2018 is to avoid the impact of the COVID-19 pandemic on the study content and to reduce the interference of external factors. On the other hand, the public data of the regional marketization index, which is an important indicator of the openness of regional development, is up to 2018, and in order to ensure the integrity of the overall research population, our research data is up to 2018. In addition, the time span of our chosen panel data is 13 years, which can meet the research needs of this paper. After robustness checks and the elimination of the endogeneity problems, our conclusions are still robust. Therefore, the time span chosen for our study data is reasonable.

6."The abbreviation in the first line in Table 2 should be used full name."

According to your suggestions, we have made adjustments in the revised draft. 

7."Is this research related to COVID-19? If the data of 2006-2018 are used, it seems no relation with COVID-19. COVID-19 broke out in the end of 2019. Thus, can this study provide any contribution or references to the post-COVID era?"

Thank you for your comments. Although the data interval we use is 2006-2018, we believe that the research in this paper is relevant to the COVID-19 pandemic and can provide an important empirical basis for the design of public policies, especially fiscal systems, in the post-COVID era. 

First, this paper introduces the research theme from the comparison of the development status of the COVID-19 pandemic. In the post-COVID era, the countries have paid significantly more attention to the construction of health care and other livelihood protection systems. China's health care system has withstood the shock of the new crown epidemic, largely thanks to the accumulation of livelihood building supported by China's fiscal system, which can improve health care coverage and accelerate economic recovery (Šteinbuka et al., 2022), thereby enhancing public welfare (Xu and Lin, 2022). Therefore, it is necessary to explore the effects of China's fiscal system on the support of livelihood security and on the quality of regional development, using China as the object of study.

Second, the research in this paper focuses closely on health care, social security employment and other infrastructures that are related to people's livelihood, which are also the key areas of fiscal concern for countries in the post-COVID era, and thus have implications for the design of public policies in various countries. This study concludes that the existing fiscal decentralization system in China has an inverted U-shaped impact on the quality of regional development. On the one hand, fiscal decentralization can increase the level of fiscal expenditure on health care, which effectively strengthens the construction of social health care system, thus effectively resisting the impact of public health security and promoting the improvement of regional development quality. On the other hand, fiscal decentralization weakens local governments' preference for social security and employment spending, which is not conducive to protecting the basic livelihood of regional residents and thus hinders high-quality regional development. In the post-COVID era, the government, which attaches more importance to the health care protection of the population, should further optimize the fiscal decentralization system and balance the construction of the health care system with the basic livelihood protection of social security employment, so as to improve the quality level of regional development as well as enhance the ability to withstand public health uncertainty shocks.

Therefore, we believe that our study can provide some implications for the design of public policies and the improvement of fiscal and taxation systems of various countries in the post-COVID era.

Reviewer 2 Report

Remarks:

- explain in more detail the selection of variables. If the variables are wrong or not appropriate, then the rest of the research paper has serious flaws,

- please explain which indicator is associated with what source of data. The current description is too general,

- the presentation of results is not clear and attractive to readers. Please add charts, graphs, etc.,

- the paper is poor regarding practical implications for international readers. Try to add more examples and recommendations in the conclusions.

Author Response

Response to Comments :

1."Explain in more detail the selection of variables. If the variables are wrong or not appropriate, then the rest of the research paper has serious flaws. "

Thank you for your comments. We explain the rationale for the selection of indicators and have added relevant content in the manuscript. The details are as follows and as shown p.7-8. And the details are as follows.

First, after the United Nations Sustainable Development Goals (SDGs) were proposed in 2015, the governments have successively put forward SDGs in line with their own development aspirations according to the stage of economic development. The Chinese government has put forward the goal of high-quality development on this basis, and has given an official definition of high-quality development: "High-quality development is development that can well meet the people's growing needs for a better life, development that reflects the new development concept, development in which innovation becomes the first driving force, coordination becomes an endogenous feature, greenness becomes the universal form, openness becomes the necessary path, and sharing becomes the fundamental purpose. " Therefore, there is a policy basis for constructing high-quality development indicators from the five dimensions of innovation, coordination, greenness, openness and sharing.

From the perspective of innovation development level, the process of innovation mainly includes multiple stages such as innovation input, innovation output and economic value of innovation results (Wang, 2021), therefore, we refer to the measurement of Wang et al. (2021), Liu (2020) and Wei and Li (2018), and construct a regional innovation development index in terms of Degree of emphasis on scientific and technological innovation, technology research and development capability and technology transformation capability to construct a regional innovation development index.

From the perspective of coordination development level, coordination development of regions includes two aspects of coordination development of industries and coordination development of regions (Wang and Chen, 2022), we further optimize the index based on the construction of coordination development index by Wang and Chen (2022), and adopt degree of rationalization of industrial structure and urban-rural income gap (Zhao and Chang, 2020).

From the perspective of green development level, the evaluation of regional green development should include Basic environmental change degree as well as development of environmental protection technology. therefore, we draw on Sun and Guo (2021), Zhao (2021) and Zhao et al. (2022) to measure the degree of basic environmental change and the number of green innovation patents to measure the development of environmental protection technology.

In terms of openness, we measure it more comprehensively in terms of the ability to attract foreign investment (Zhou et al., 2022), the level of regional marketization (Hua et al., 2021), and the Internet penetration rate (Chen et al., 2022).

In terms of shared development, shared development is reflected in an overall increase in social welfare levels, which we measure using public consumption levels (Dai et al., 2022) and basic social security indicators such as educational attainment, healthcare, and social security and employment, thus reflecting the shared nature of economic development.

Second, we use TOPSIS to measure the quality development index, which can effectively reduce the measurement error by objectively weighting the index to avoid the influence of subjective factors of principal component analysis (PCA) and other measurements.

Third, the evaluation analysis of the results of the regional development quality index in the manuscript shows that the regional variability of our constructed quality development index is basically consistent with the degree of regional economic development, which indicates the reliability of our constructed index.

2."Please explain which indicator is associated with what source of data. The current description is too general."

Thank you for your comments. We have added relevant content to the data sources section as shown p.10-11. And the details are as follows:

On the one hand, the reform of fiscal expenditure items after 2006, so we need to ensure the consistency of the statistical caliber of the data. And the Eleventh Five-Year Plan in 2006 clearly pointed out that "adhering to the people-oriented approach, changing the development concept, innovating the development model, improving the quality of development, implementing the "five coordination", and effectively turn economic and social development into the track of comprehensive, coordinated and sustainable development." The focus of economic and social development has begun to tilt toward innovation-driven and high-quality development, emphasizing comprehensive and sustainable development. On the other hand, we need to exclude the impact of the COVID-19 pandemic on economic development since 2019 in the design of our study, and the marketability index, an important indicator of the openness of regional development, is publicly available as of 2018. Therefore, to ensure the integrity and reliability of the data, we use the relevant data for 30 provincial regions (excluding Tibet, Hong Kong, Macao and Taiwan) from 2006-2018.

The data sources we used are listed as follows. First, we have several sources for the data involved in the construction process of regional quality development indicators. Specifically, the original data of degree of emphasis on scientific and technological innovation (Z11) and technology research and development capability (Z12) are obtained from the China Science and Technology Statistical Yearbook. The original data of technology transformation capability (Z13) are obtained from the China High Technology Industry Statistical Yearbook and the database of the National Bureau of Statistics. The level of coordinated development of regional industries (Z21) is measured by the Thiel index. The original data of PM2.5 is obtained from the global raster data of Washington University in St. Louis, USA. The number of green patents granted in the region (Z35) is based on the number of green invention patents granted to listed companies in the patent database of the China National Intellectual Property Administration. The marketization level index (Z42) is obtained from the China Market Index Database. The original data of the remaining indicators are obtained from the database of the Chinese National Bureau of Statistics. And among the three-level indicators, Z21 and Z31 are negative indicators, while the rest are positive indicators. In addition, the original data of the independent variable, control variables and mediating variables are obtained from the China Statistical Yearbook and the database of the Chinese National Bureau of Statistics.

3."The presentation of results is not clear and attractive to readers. Please add charts, graphs, etc. "

Thank you for your comments, we have added some graphics to the results evaluation for additional explanation as shown p.12.

4."The paper is poor regarding practical implications for international readers. Try to add more examples and recommendations in the conclusions."

Thank you for your comments. We have revised the content of the conclusion section as shown in p.18-19. 

Reviewer 3 Report

1. The present manuscript deals with economic aspect of sustainable development through fiscal decentralization in China. Despite the timeliness of the paper and its scientific merit, it is not fit for publication in the journal Land. The main reason for the same is that  the present manuscript does not meet the Aim of the journal. Similarly, the study does not come under any of the subject areas of the journal. In my opinion, the manuscript is more suitable for Sustainability science journals.

2. While revising the manuscript, the authors should include the aspects of United Nations sustainable development goals that can be achieved through applying the approach of fiscal decentralization.

Author Response

Commnets :

1." The present manuscript deals with economic aspect of sustainable development through fiscal decentralization in China. Despite the timeliness of the paper and its scientific merit, it is not fit for publication in the journal Land. The main reason for the same is that the present manuscript does not meet the Aim of the journal. Similarly, the study does not come under any of the subject areas of the journal. In my opinion, the manuscript is more suitable for Sustainability science journals."

Thank you for your acknowledgement of the research content of this manuscript and your suggestions for matching it to the journal.

On the one hand, the study is in line with the research theme of the target special issue. The special issue is "the result of growing concern for happiness and social wellbeing. And it will build from the different contributions to physical and mental health (eventually, happiness) into the target goals of projecting the future based on meaningful public policies." The manuscript explores the relationship between fiscal decentralization, expenditure preferences in livelihood categories and regional quality development. And it confirms that the improvement of the fiscal decentralization system can influence the government's preference for welfare spending in the areas of health care and social security and employment, improve the public's lifestyle and quality, and thus influence the quality of regional development and welfare level.

On the other hand, the special issue will analyze the possible public policies directed to welfare solutions in an urban context. And our manuscript is based on the social goal of achieving high-quality development, and explores possible ways to increase the level of health protection and welfare for the people's livelihood from the perspective of improving the government's fiscal and taxation system. This is in line with the research theme of the special issue.

Therefore, we believe that our manuscript may be compatible with the theme of this special issue study in LAND.

2."While revising the manuscript, the authors should include the aspects of United Nations sustainable development goals that can be achieved through applying the approach of fiscal decentralization."

Thank you for your comments. The quality development goals are based on the United Nations Sustainable Development Goals (SDGs) and they are closely linked to each other. As you suggested, we have added related sections in the introduction, explanatory variables formulation, and other sections. as shown on pages 2, 3, and 6 of the paper. 

Thanks again for your comments, and if you have questions we'd be happy to address them.

Round 2

Reviewer 1 Report

The authors fully respond to my comments.

Reviewer 2 Report

Accept in present form.

Reviewer 3 Report

The authors have provided valid justifications for the comments. Thus, the manuscript can be accepted in its current form.